# Anomalous diffusion analysis of semantic evolution in major Indo-European languages

**Bogdán Asztalos**[1], **Gergely Palla**[1,2]*, **Dániel Czégel**[3,4]

**1** Deptartment of Biological Physics, Eötvös University, Budapest, Hungary, **2** Health Services Management Training Centre, Semmelweis University, Budapest, Hungary, **3** Institute of Evolution, Centre for Ecological Research, Budapest, Hungary, **4** Parmenides Center for the Conceptual Foundations of Science, Pöcking, Germany

* palla.gergely@emk.semmelweis.hu

**Data Availability Statement:** Data is downloaded from Google Books Ngrams Viewer Database, version 2, available at https://storage.googleapis.com/books/ngrams/books/datasetsv2.html. Our

## Abstract

How do words change their meaning? Although semantic evolution is driven by a variety of distinct factors, including linguistic, societal, and technological ones, we find that there is one law that holds universally across five major Indo-European languages: that semantic evolution is subdiffusive. Using an automated pipeline of diachronic distributional semantic embedding that controls for underlying symmetries, we show that words follow stochastic trajectories in meaning space with an anomalous diffusion exponent $\alpha = 0.45 \pm 0.05$ across languages, in contrast with diffusing particles that follow $\alpha = 1$. Randomization methods indicate that preserving temporal correlations in semantic change *directions* is necessary to recover strongly subdiffusive behavior; however, correlations in change *sizes* play an important role too. We furthermore show that strong subdiffusion is a robust phenomenon under a wide variety of choices in data analysis and interpretation, such as the choice of fitting an ensemble average of displacements or averaging best-fit exponents of individual word trajectories.

## Introduction

Cumulative cultural evolution at enormous scale and speed makes us a strikingly different species than the rest of the living world [1]. The ability of accumulating techniques and solutions one bit at a time, aggregating them across time and space gave us unprecedented power, that we use to shape the world. But how does this process of human cultural evolution unfold? Are there universal patterns that hold across cultures and eras? One human activity that allows us to zoom into the cumulative patterns of human thought is language production. We all comprehend and produce, all the time, keeping the cogwheels of language evolution moving. How do these cogwheels move?

The study of human language evolution necessitates an interdisciplinary approach and has significant theoretical and practical ramifications in a variety of fields from the core linguistic subfield of etymology to the more computational areas, like natural language processing [2–4]. Mathematical models, simulations, and controlled experiments suggest that several factors play a role in semantic change, including the efficiency of communication as well as cognitive

Python code is available at https://github.com/abogdan271/histwords.

**Funding:** This project has received funding from the European Union's Horizon 2020 research and innovation programme under grant agreement no. 101021607 and was partially supported by the National Research, Development and Innovation Office under grant no. K128780 and by the the European Union project RRF-2.3.1-21-2022-00004 within the framework of the Artificial Intelligence National Laboratory. The funders had no role in study design, data collection and analysis, decision to publish, or preparation of the manuscript.

**Competing interests:** The authors have declared that no competing interests exist.

pressures on language acquisition [5, 6]. In the case of empirical quantitative investigations, recent efforts focused on uncovering universal (i.e. language, time, genre, etc. independent) dynamical rules that govern *frequency changes* of (variants of) words or ordered combinations of them, called n-grams [7]. This effort provided us a handful of remarkable discoveries, including peaks or valleys of individual terms mirroring specific societal-political processes (e.g., censorship, propaganda, ideology shifts, cultural-technological drift, natural or social catastrophes, etc.), aggregate behavior of groups of terms (e.g., decay rate of fame of a cohort of people across historical time) [8], competition dynamics among linguistic variants (e.g., [9]) or synonyms, and even a marked difference between the temporal correlational patterns of word frequencies referring to natural or social processes [10].

N-gram frequencies also serve as the basis of the automated estimation of semantic similarities between words. Building on the distributional hypothesis, paraphrased as "a word is characterized by the company it keeps" [11]: similarity in meaning can be approximated by comparing neighborhoods of words over large corpora [12]. Many flavors of the distributional hypothesis have been formalized with classical methods [13, 14], yet the advent of large-scale estimation of semantic similarity came with the "machine learning revolution" in the 2010s [15]. From these techniques, a successful and computationally efficient variant is the Word2-vec embedding algorithm [16–18]. Instead of calculating the vector representation of words explicitly, Word2vec solves a prediction problem by estimating semantic similarity based on sampling co-occurrences of words. This implements dimension reduction over the set of pairwise semantic similarities to embed words in a relatively low dimensional space (e.g., tens of thousands of words in a few hundred dimensional Euclidean space) [19]. With a corpus of time-labeled co-occurrences in hand, one can, in principle, track changes in these semantic similarities in an automated way by comparing embeddings corresponding to different times.

Indeed, a first endeavor utilizing this approach has identified two novel statistical patterns governing the *semantic change* of words: the law of conformity, stating that words with lower frequency change their meaning faster, and the law of innovation, finding that more polysemous words also tend to exhibit higher rate of semantic change [20]. Although the authors applied multiple word embedding variants, the validity of these results mirroring social-technological-linguistic effects as opposed to being mathematical artifacts is still debated [21]. It is because although state-of-the-art word embedding methods match reported semantic similarities across experiments, languages, and training corpora, they produce systematic biases over non-semantic features. One particular bias is due to the power-law distribution of word occurrences, known as Zipf's law [22, 23], resulting in embeddings where low-frequency words tend to appear close to the center of the embedding whereas high-frequency words are pushed to the periphery. Such implicit biases point to the necessity of careful comparison of obtained results with those based on various randomized replicas of the dataset, systematically removing statistical dependencies until the phenomena at hand is no longer apparent.

Furthermore, with *diachronic* embeddings, three additional issues arise. First, embedding dimensions are arbitrary. There is no guarantee that dimensions match across subsequent timesteps. Second, Word2vec is a sampling-based method, and therefore, it is non-deterministic. Third, there are underlying symmetries along which embeddings are degenerate: namely, a set of transformations that change embeddings (and the embedding of context words) yet leave co-occurrences invariant. In order to tackle all four aforementioned obstacles, in this paper we develop a dynamical alignment method that is i) symmetry-agnostic, ii) averages over many runs to yield a robust estimate of embedding positions and their variances, iii) based on these principles, finds the best alignment over all timesteps, and iv) compares obtained results with those of systematically randomized versions of input data, removing various statistical dependencies in a step-by-step manner. Current diachronic embedding methods

mostly focus on point iii) [24–26]; here we suggest that all points above are needed for a robust estimation of semantic trajectories.

Starting from word trajectories, we construct the ensemble of all trajectories and ask whether such an ensemble of trajectories obeys any robust statistical regularities. We focus on measuring the systematic deviation of such trajectories from those of standard diffusing particles (i.e., a random walk), quantified by the *anomalous diffusion exponent α*, defined as [27–30]

$$|\Delta x|^2 \sim t^\alpha \tag{1}$$

where $\Delta x = x(t) - x(0)$ is the displacement of the position vector $x$ of a word in the $D$-dimensional embedding space at time $t$ compared to its starting position at time $t = 0$.

Standard diffusion, corresponding to random walk-like trajectories, is characterized by an anomalous diffusion exponent $\alpha = 1$. There are, however, a variety of more general microscopic rules that generate trajectories that still belong to the $\alpha = 1$ class, including i) trajectories with exponentially decaying memory, ii) the collective motion of weakly interacting particles, iii) varying step sizes or waiting times between steps unless those are not power-law distributed, iv) random walk in not very statistically heterogeneous media. Consequently, the diffusive ($\alpha = 1$) class is very general, and potential deviations from it indicate the violation of i), ii), iii), or iv) above, or other characteristics that are not discussed here. In other words, an observed anomalous diffusion exponent $\alpha \neq 1$ implies (a combination of) particular underlying microscopic rules that generate the dynamics, notably power-law decaying memory (violation of i) above, modeled by e.g., fractional Brownian motion), the collective motion of strongly interacting particles, such as in "jamming" (violation of ii)), power-law distributed step sizes or waiting times (violation of iii)), diffusion in disordered, such as self-similar, media (violation of iv)), or changing dynamical rules, such as aging [27–30]. This makes the anomalous diffusion exponent $\alpha$ a simple yet powerful indicator of the possible underlying microscopic statistical rules that govern the unfolding of the process, in this case, the meaning change of words. Anomalous diffusion is further subdivided into superdiffusion and subdiffusion, with $\alpha > 1$ and $\alpha < 1$, respectively, with different possible underlying generative mechanisms.

Deviation from diffusing behavior is not an unknown phenomenon: as an example, subdiffusive behavior has been observed in various within-cell processes, such as in the stochastic trajectory of messenger RNA inside living E. coli cells, $\alpha \approx 0.7$, [31], channel proteins in the membranes of living cells, $\alpha \approx 0.9$, [32], and telomeres within eucaryotic cell nuclei, $\alpha \approx 0.3$, [33], indicating the statistical structure of the ambient space or the type of active motion by the macromolecules themselves. Fig 1d illustrates subdiffusive trajectories, corresponding to various $\alpha < 1$ exponents, generated by fractional Brownian motion (fBm) [34]. Note that we chose fBm for visualization purposes only; fBm generates trajectories with long-range temporal correlations, corresponding to the violation of point i) above.

Actual semantic trajectories might be governed by a mixture of underlying dynamical rules, as discussed above. Measuring the anomalous diffusion exponent $\alpha$ classifies the semantic change in the context of stochastic processes, and helps to explore the possible stochastic dynamical rules that drive linguistic change.

## Methods

### Corpus

To construct semantic trajectories of words, we use the downloadable version of the Google Books Ngram Viewer database version 2, downloaded from https://storage.googleapis.com/

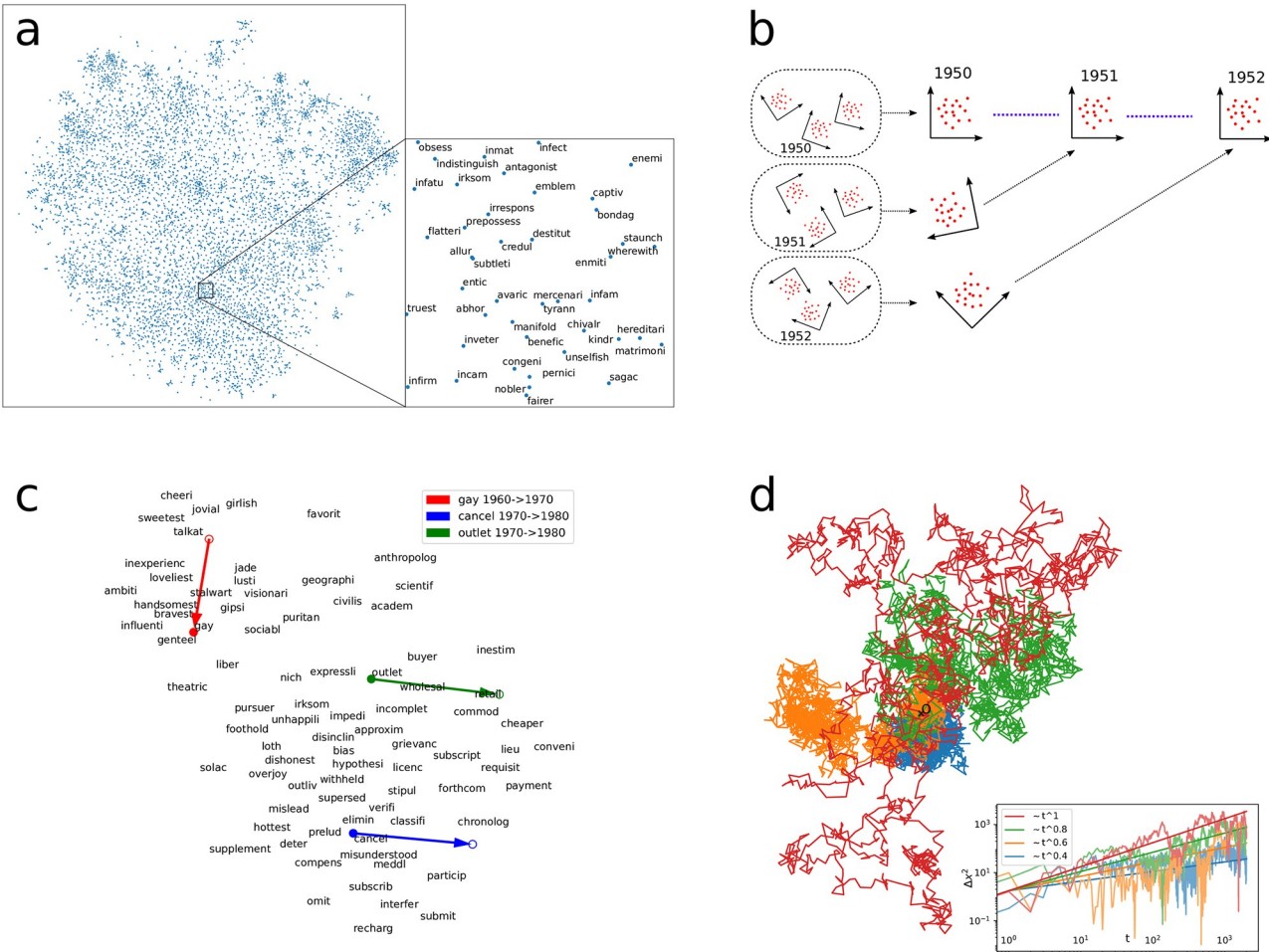

**Fig 1. Illustration of word embedding, semantic change, and subdiffusive trajectories.** (a) 2D projection of a high-dimensional word embedding, illustrating semantic similarity. 300D embedding of lemmas in the Google ngram English fiction database are generated by Word2vec Skip-gram model and are nonlinearly projected to 2D using t-sne [35]. (b) Alignment of embeddings. First, multiple embeddings of the same year are aligned and averaged to reduce embedding noise, then, these averaged embeddings are aligned across time to achieve maximally smoothened word trajectories. (c) The resulting diachronic embedding makes it possible to visualize semantic change. Three selected words (gay, cancel, outlet) with most change within a decade are shown with respect to the time averaged position of other semantically related words. (d) Illustration of subdiffusive trajectories generated by fractional Brownian motion. Inset: mean squared distance scales as $(\Delta x)^2 \sim t^\alpha$ with anomalous diffusion exponent $\alpha$.

books/ngrams/books/datasetsv2.html, as time-labelled corpora available in multiple languages [8, 36]. In this work, we included five languages from the Indo-European family with the most available data, English, French, German, Spanish, and Italian. Notable limitations of the Google Ngram corpus include the lack of meta-data, and correspondingly, the equal importance assigned to a large variety of sources, including those by non-native speakers. As a special case, the English dataset suffers from significant overrepresentation of academic literature [37, 38]; in order to correct for this bias, we use the *English fiction* corpus as a representation of English [37].

## Language processing

We used the snowball stemmer of the natural language toolkit (nltk) module of Python [39] to lemmatize words, and then we filtered out stopwords [40], before collecting all co-occurrences

within a window size 2, as suggested by Levy et al. [15] and Hamilton et al. [20], illustrated by Fig 2a–2c.

## Temporal grouping

Since temporal annotation of ngram occurrences follow a yearly resolution, we first collected co-occurrences by year, and then we applied a 10-year sliding window to increase vocabulary size and to smoothen data. Although setting the sliding window size to 10 years is an arbitrary choice, we show in S1 File (namely in S5 Fig in S1 File) that as window size increases gradually from 1 year to 10 years, the exponent $\alpha$ approximately converges to its value at 10 years. We then filtered out words that do not appear at least 100 times in each time window.

While Google's raw database initially provides ngram data from almost all years in the past few centuries, the number of occurrences decreases significantly as we look further back in time. This is a crucial factor in our methodology, since a certain amount of co-occurrence statistics is required for the embedding model to find a proper representation for the word meanings. Since the minimum number of words known by a native speaker is roughly 10 000 [41], we prescribed that we have a vocabulary with at least that size in each time step by only keeping data from years after 1950. If we went back further in time, we would have had a much smaller set of words to work with. This process is shown in Fig 2d and 2e.

## Semantic embedding

For constructing word embeddings, we use Word2vec's Skip-gram model with negative sampling (SGNS), which is one of the most widely used word embedding algorithms due to its computational efficiency and ability to capture semantic relationships in a simple mathematical form [16, 42]. It uses a two-layer neural network to represent each word $i$ with two $D$-dimensional vectors called 'word vector' ($v_i$) and 'context vector' ($w_i$) such that a global cost function $C$, depending on all word vectors and all context vectors is (approximately) minimized. The objective of the Skip-gram model is to optimize the estimated empirical log-probabilities

$$\log p(j \mid i) \propto v_i \cdot w_j \tag{2}$$

of word $j$ occurring in the context of word $i$, using the cost function $C$ defined as

$$C = \frac{1}{L} \sum_{i=1}^{L} \sum_{j \in \mathcal{C}(i)} \log p(j \mid i), \tag{3}$$

where $L$ is the length of the text and $\mathcal{C}(i)$ is the linguistic context of word $i$. Following the recommendations of [15, 20], we set the dimension $D$ of the embedding space to $D = 300$ and the context size to 2. Fig 1a visualizes a single embedding projected to 2 dimensions by t-sne [35], a non-linear dimension reduction method.

## Measuring semantic distance

In this paper, we self-consistently define semantic distance $|x - y|$ between vectors $x$ and $y$ in the D-dimensional embedding space as Euclidean distance, $|x - y|^2 = \sum_{i=1}^{D} (x_i - y_i)^2$. This is a departure from the most commonly used semantic similarity measure, cosine similarity $|x - y|_{\cos}$, which is favored because it accounts for the rescaling symmetry, $x \to \lambda x$, by setting $|\lambda_1 x - \lambda_2 y|_{\cos} = |x - y|_{\cos}$ [13, 42]. We argue, however, that finding the most general class of underlying symmetries and measuring distance separately is a more principled procedure, especially

**Fig 2. Key steps of the pipeline that generates diachronic embeddings from google ngram data.** (a) Lemmatization of words. (b) Filtering stopwords. (c) Counting co-occurrences with context window of length 2. (d) Smoothening co-occurrences by applying a sliding time window to co-occurrence counts. (e) Vocabulary, i.e., the final set of lemmas we include in our analysis, is defined by the lemmas that occur at least 100 times at all time windows. (f) Training dataset was created by subsampling all co-occurrences in order to have a constant number of co-occurrences $N_c = 10^7$ at each time window. (g) Word embedding by Word2vec with Skip-gram. (h) Alignment of different embeddings (both within and across years). (i) Randomization of the temporal order of step *sizes* of each word trajectory separately. (j) Randomization of the temporal order of step *directions* of each word trajectory separately.

when semantic distances have numerical and not just ordinal meaning. In particular, we choose Euclidean distance for the following reasons.

1. In the Skip-gram model, the difference of two word vectors, $v_i - v_j$, has semantic meaning. By interpreting the space of context words as a dual space (see S1 File for details), the meaning of $v_i - v_j$ can be understood by projecting it to various context words, akin to the definition of linear functionals in dual space,

$$
\begin{aligned}
(v_i - v_j) \cdot w_k \ &= v_i \cdot w_k - v_j \cdot w_k \\
&\propto \log p(k \,|\, i) - \log p(k \,|\, j) \\
&\propto \log \left( \frac{p(k \,|\, i)}{p(k \,|\, j)} \right).
\end{aligned}
\tag{4}
$$

Since this result is valid for any context word $k$, $v_i - v_j$ measures the relative log-probability of word $i$ versus word $j$ occurring in the context of word $k$, for all contexts $k$. This is in line with the starting hypothesis of distributional semantics, which states that two words have a similar meaning if their co-occurrence statistics are similar.

2. Euclidean distance is a *metric*, obeying triangle inequality, commutativity, and other axioms, whereas other measures, such as cosine similarity are not. This is important when semantic trajectories are interpreted as trajectories in metric spaces, for example, by measuring their squared displacement over time, $|\Delta x|^2(t) = |x(t) - x(t = 0)|^2$.

3. As discussed above, embedding dimensions are arbitrary. They need to be aligned for meaningful comparison across multiple embeddings (regardless of whether it coming from the same co-occurrence data or a different one). Here we follow the most commonly used method, called the Procrustes algorithm [43], that finds an orthogonal transformation that minimizes the total Euclidean distance between corresponding points in the two embeddings. Measuring semantic distance as Euclidean distance is consistent with this alignment method.

4. As a consequence of keeping the size of the word cloud constant, defined by Eq (11), two embeddings belong to the same equivalence class if they can be orthogonally transformed to each other, discussed above and shown below in Methods. Orthogonal transformations are defined by keeping Euclidean distance, also consistent with our definition.

## Diachronic embedding

We first subsample the co-occurrences corresponding to each time window to eliminate systematic sample size bias (the amount of data included in the Google Ngram database steadily increases with time in all five languages) as shown by Fig 2f. We set this sample size to $N_c = 10^7$ co-occurrences, set by the first (few) time windows with the least amount of data. We observe that $N_c = 10^7$ sets a good tradeoff between sample size (and thus trajectory noise), vocabulary size, and trajectory length for this database. As shown in S1 File (namely in S1 Fig in S1 File), using larger samples will not give us embeddings with more information. This is because increasing the number of input co-occurrences only makes the word cloud larger without altering its structure.

After generating the $D = 300$ dimensional embeddings for each time window $M = 80$ times by Word2vec with Skip-gram, we first align all $M$ embeddings within a time window. Then,

word positions across embeddings are averaged to obtain a more robust estimate of semantic positions for each word at a given time window. These average embeddings, one for each time window, are then aligned across time, as shown by Fig 2g and 2h.

The alignment of two embeddings consists of two steps. First, one has to define the class of transformations that keep semantic relationships invariant, and second, the transformation within this class that minimizes the difference between the two embeddings has to be selected. For the first step, we find (see below for a proof) that the requirement of keeping the total size of the word cloud constant, as defined by Eq (11), restricts all linear transformations to those that are orthogonal. This is also in line with both measuring semantic distance as Euclidean distance, as discussed above, and also with the most commonly used alignment method called the orthogonal Procrustes method, which finds that the best transformation $R$ that accounts for this rotational (orthogonal) freedom is given by

$$R = \arg \min_{Q \in O(D)} ||W'Q - W||_F \tag{5}$$

where the rows of the $N \times D$ matrix $W'$ and $W$ contain the word vectors of the aligned and reference embedding, respectively, and $||\cdot||_F$ is the Frobenius matrix norm. We use the analytical solution [43] for $R$ to find optimal alignments both within a time window and among time windows.

## Constant word cloud size defines an orthogonal embedding symmetry

Word cloud size, defined by Eq (11), is written as

$$\text{Tr } V(W) = \sum_{j=1}^{D} \text{Var}_i \left( W_{ij} \right) \tag{6}$$

where $V(W)$ is the empirical covariance matrix of word positions, extracted from $W$, the $N \times D$ matrix with its rows corresponding to the $D$-dimensional word vectors of all $N$ words, and $\text{Var}_i (W_{ij})$ is the variance of the word positions along dimension $j$. Using the definition of variance,

$$
\begin{aligned}
\text{Tr } V(W) &= \sum_{j=1}^{D} \text{Var}_i \left( W_{ij} \right) = \sum_{j=1}^{D} \left[ \frac{1}{N} \sum_{i=1}^{N} W_{ij} W_{ij} - \left( \frac{1}{N} \sum_{i=1}^{N} W_{ij} \right)^2 \right] \\
&= \frac{1}{N^2} \sum_{j=1}^{D} \sum_{i=1}^{N} \left[ N W_{ij} W_{ij} - W_{ij} \sum_{k=1}^{N} W_{kj} \right].
\end{aligned}
\tag{7}
$$

The size of the transformed word cloud $W'$, defined as $W' = RW$ is

$$
\begin{aligned}
\text{Tr } V(W') &= \frac{1}{N^2} \sum_{j=1}^{D} \sum_{i=1}^{N} \left[ N \sum_{l=1}^{D} W_{il} R_{lj} \sum_{m=1}^{D} W_{im} R_{mj} - \sum_{l=1}^{D} W_{il} R_{lj} \sum_{k=1}^{N} \sum_{m=1}^{D} W_{km} R_{mj} \right] \\
&= \frac{1}{N^2} \sum_{j,l,m=1}^{D} \sum_{i=1}^{N} R_{lj} R_{mj} \left[ N W_{il} W_{im} - W_{il} \sum_{k=1}^{N} W_{km} \right].
\end{aligned}
\tag{8}
$$

The difference between $\text{Tr } V(W)$ and $\text{Tr } V(W')$ is

$$\text{Tr } V(W) - \text{Tr } V(W') = \frac{1}{N^2} \sum_{l,m=1}^{D} \sum_{i=1}^{N} \left( \delta_{ml} - \sum_{j=1}^{D} R_{lj} R_{mj} \right) \left[ N W_{il} W_{im} - W_{il} \sum_{k=1}^{N} W_{km} \right]. \tag{9}$$

The transformation $R$ satisfies the requirement of constant cloud size Tr $V(W)$ − Tr $V(W')$ = 0 only if

$$\sum_{j=1}^{D} R_{lj}R_{mj} = \delta_{ml}, \tag{10}$$

also written in matrix form as $RR^T = \mathbf{1}$, i.e., $R$ needs to be orthogonal.

## Mean versus ensemble average anomalous diffusion exponent

As explained in the Introduction, distributional semantics does not directly define the meanings of words; it treats them as statistical properties based on their contexts. Studying word meanings in this framework only makes sense in relation to other words, which are also defined relationally. This means that one cannot extract all the semantic information of an individual word from its embedded trajectory alone, but other word trajectories also have to be considered, and their collective behaviour should be studied. When dealing with measurable quantities, such as anomalous diffusion exponents, this requires analyzing their distribution and averages.

Averages of anomalous diffusion exponents corresponding to single-word trajectories are calculated in two different ways. The *mean anomalous diffusion exponent* $\bar{\alpha}$ is computed by first fitting an anomalous diffusion exponent $\alpha_i$ to the trajectory of each word $i$, and then averaging the fitted exponents, $\bar{\alpha} = \sum_{i=1}^{N} \alpha_i/N$, where $N$ is the total number of words. To compute the *ensemble average anomalous diffusion exponent*, we first average $|\Delta x_i(t)|^2$, the squared displacement of word $i$ over time to obtain the mean squared displacement over time, $|\Delta x(t)|^2 = \sum_{i=1}^{N} |\Delta x_i(t)|^2/N$, and then we fit an anomalous diffusion exponent to $|\Delta x(t)|^2$, called ensemble average anomalous diffusion exponent $\langle\alpha\rangle$.

## Trajectory randomization methods

Fig 2i and 2j illustrate the two types of trajectory randomization methods we use in this paper to generate the results shown by Fig 3c and 3d and Table 1: randomization of step sizes and randomization of step directions. Within each type, "randomization" refers to drawing step sizes and directions from various distributions constructed from the original trajectories, described as follows.

*Random sizes*: step sizes were sampled from a normal distribution with mean and standard deviation corresponding to that of the step sizes of the original trajectory; *sizes from distribution*: step sizes were sampled from the set of all step sizes corresponding to all words; *shuffled sizes*: step sizes were sampled from the set of step sizes corresponding to the same word; in other words, the temporal order of step sizes of a trajectory has been shuffled; *original sizes*: step size for every word at every time step was set to its original value.

*Random directions*: directions were sampled uniformly over a D-dimensional sphere; *shuffled directions*: the temporal order of step directions of a trajectory has been shuffled; *original directions*: the direction for every word at every time step was set to its original value.

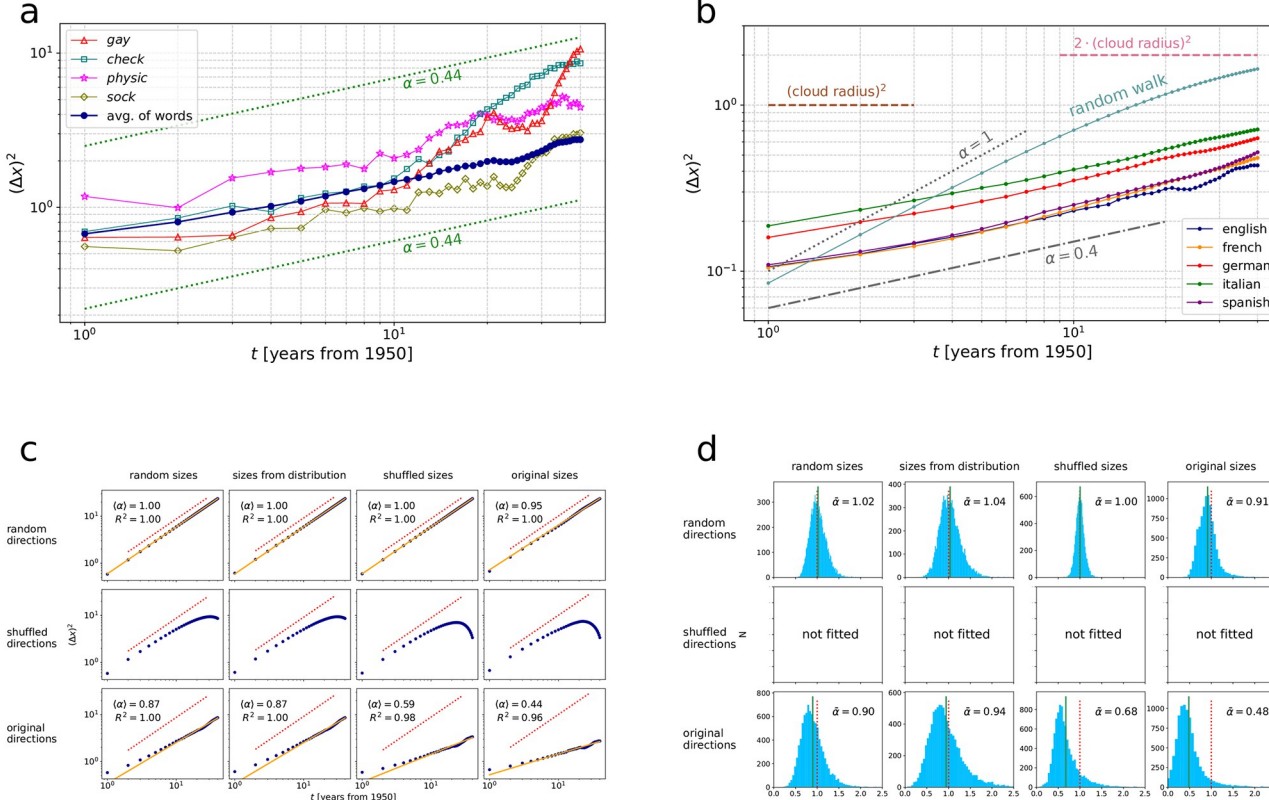

**Fig 3. Anomalous diffusion analysis of semantic trajectories.** (a) Squared displacement $(\Delta x)^2$ of selected English words over time, along with average squared displacement over all words, shown in blue, which can be approximated by $\langle (\Delta x)^2 \rangle \sim t^{\langle \alpha \rangle}$ with ensemble-average anomalous diffusion exponent $\langle \alpha \rangle \approx 0.44$. (b) Average squared displacement of words over time in five languages, English, French, German, Italian, and Spanish. This is compared to an ensemble of random walkers under the same constraint given by a constant total word cloud size, defined by Eq (11). (c) Average squared displacement $\langle (\Delta x)^2 \rangle$ of English words under various combinations of trajectory randomization methods. Columns and rows differ in the way step *sizes* and *directions* are randomized, respectively, see details in text. Keeping the original sequence of step directions is necessary to recover strongly subdiffusive behavior (bottom row), but not sufficient: the more information regarding step sizes is kept, the more its best-fit diffusion exponent approaches its original value of $\langle \alpha \rangle \approx 0.44$. (d) Distribution of diffusion exponents $\alpha$ fitted to the trajectory of each word separately. Their average $\bar{\alpha}$ is close to the ensemble averages $\langle \alpha \rangle$ shown in c.

## Results

### Maximally representation-agnostic temporal embedding

While constructing semantic trajectories of words from time-labelled co-occurrence data, one needs to account for two sources of arbitrariness in the process: stochasticity and symmetry. The first, stochasticity, comes from the nature of modern data-efficient semantic embedding methods, such as Word2vec. Since these approaches use a neural network to find the best possible embedding, sampling techniques, initial weights, and the choice of numerical optimization method (such as stochastic gradient descent) lead to non-deterministic embeddings. Second, the cost $C$ itself exhibits multiple global minima associated with the same input data. In particular, the form of the estimated log-probabilities defined in (2) implies that any transformation that leaves all dot products $v_i \cdot w_j$ invariant results in the exact same cost $C$. The simplest example of such transformation is a linear rescaling of word vectors $v \mapsto \lambda v$ and inverse rescaling of context word vectors $w \mapsto \lambda^{-1} w$, but in principle, any invertible linear transformation of word vectors $v \mapsto Rv$, together with (the transpose of) its inverse applied to context vectors $w \mapsto (R^{-1})^T w$ leaves the dot product invariant (see Methods for details).

**Table 1. Ensemble-average anomalous diffusion exponent $\langle\alpha\rangle$ and the average $\bar{\alpha}$ of the anomalous diffusion exponents of all words, under various combinations of trajectory randomization methods, for all five languages.** Grey cells show the exponents of the original, non-randomized trajectories. Note that the exponents for English are extracted from Fig 3c and 3d; for the other four languages, the analogous Figures are included in S3 and S4 Figs in S1 File.

| ENGLISH | random directions | | original directions | |
|---|---|---|---|---|
| | $\langle\alpha\rangle$ | $\bar{\alpha}$ | $\langle\alpha\rangle$ | $\bar{\alpha}$ |
| random sizes | 1.00 | 1.02 | 0.87 | 0.90 |
| sampled sizes | 1.00 | 1.04 | 0.87 | 0.94 |
| shuffled sizes | 1.00 | 1.00 | 0.59 | 0.68 |
| original sizes | 0.95 | 0.91 | **0.44** | **0.48** |
| FRENCH | random directions | | original directions | |
| | $\langle\alpha\rangle$ | $\bar{\alpha}$ | $\langle\alpha\rangle$ | $\bar{\alpha}$ |
| random sizes | 1.00 | 1.03 | 0.90 | 0.93 |
| sampled sizes | 1.00 | 1.05 | 0.90 | 0.98 |
| shuffled sizes | 1.00 | 1.00 | 0.57 | 0.68 |
| original sizes | 0.98 | 1.02 | **0.49** | **0.64** |
| GERMAN | random directions | | original directions | |
| | $\langle\alpha\rangle$ | $\bar{\alpha}$ | $\langle\alpha\rangle$ | $\bar{\alpha}$ |
| random sizes | 1.00 | 1.02 | 0.81 | 0.84 |
| sampled sizes | 1.00 | 1.03 | 0.81 | 0.87 |
| shuffled sizes | 1.00 | 1.00 | 0.51 | 0.57 |
| original sizes | 0.97 | 0.98 | **0.41** | **0.48** |
| ITALIAN | random directions | | original directions | |
| | $\langle\alpha\rangle$ | $\bar{\alpha}$ | $\langle\alpha\rangle$ | $\bar{\alpha}$ |
| random sizes | 1.00 | 1.02 | 0.77 | 0.79 |
| sampled sizes | 1.00 | 1.02 | 0.78 | 0.81 |
| shuffled sizes | 1.00 | 1.00 | 0.51 | 0.55 |
| original sizes | 0.94 | 0.96 | **0.40** | **0.44** |
| SPANISH | random directions | | original directions | |
| | $\langle\alpha\rangle$ | $\bar{\alpha}$ | $\langle\alpha\rangle$ | $\bar{\alpha}$ |
| random sizes | 1.00 | 1.03 | 0.93 | 0.97 |
| sampled sizes | 1.00 | 1.06 | 0.93 | 1.02 |
| shuffled sizes | 1.00 | 1.01 | 0.63 | 0.78 |
| original sizes | 0.93 | 0.94 | **0.49** | **0.66** |

However, an additional constraint comes from focusing solely on words (and not contexts) when constructing temporal trajectories: the constraint that ensures that the ambient embedding space does not shrink or expand over time, formalized as

$$\mathrm{Tr}\, V = \sum_{i=1}^{D}\sigma_i^2 = \mathrm{const},$$ (11)

where $V$ is the $D \times D$ empirical covariance matrix of word positions (with $D = 300$ being the embedding dimension) and $\sigma_i$ is the standard deviation of word positions along principal dimension $i$. As shown in Methods, this reduces the possible transformations $R$ to the orthogonal ones, obeying $R^{-1} = R^T$. Consequently, a single embedding is identified with an equivalence class containing $\{Rv_i, Rw_j\}$ with any orthogonal $R$.

We use this orthogonal freedom to define maximally representation-agnostic trajectories in two steps as shown in Fig 1b. First, we minimize the effect of stochasticity, described below, in

any single time. Without stochasticity, any embedding starting from the same co-occurrence data would belong to the same equivalence class, and therefore it would be possible to find a transformation $R_j$ for each embedding $j$ such that they all numerically coincide. With stochasticity, however, different embedding realizations, starting from the same data, land in (slightly) different equivalence classes, and as a consequence, perfect alignment among them is not possible. The best one can do is to find a transformation for each embedding such that an overall distance measure between all embeddings is minimized (see Methods for details). This allows us to "average out", i.e., minimize the effect of stochasticity at any single timestep by taking the average of all aligned embeddings. Second, we align averaged embeddings corresponding to *different* times in the same way: we find an orthogonal transformation that minimizes the distance between the embedding at subsequent times to construct word trajectories. Such word trajectories are thus maximally smoothened. Although this raises the question of whether maximal smoothening washes away real phenomena, we will see that this smoothening procedure applied to random walk does not change measured observables such as the anomalous diffusion exponent $\alpha$ of the process. Fig 1c shows the measured semantic change of three selected words within a decade, projected to 2 dimensions, visualized over a static background.

## Semantic subdiffusion across languages

As detailed in Methods, to measure the actual value of the anomalous diffusion exponent $\alpha$, we need to take averages over the whole vocabulary. We proceed from single trajectories to an ensemble of trajectories in two alternative ways: (i) we first average $|\Delta x|^2(t)$ over individual trajectories to obtain $\langle |\Delta x|^2(t) \rangle$ and then we fit the *ensemble-average anomalous diffusion exponent*, $\langle \alpha \rangle$, based on $\langle |\Delta x|^2(t) \rangle \sim t^{\langle \alpha \rangle}$; (ii) alternatively, we first fit the anomalous diffusion exponent $\alpha$ to single trajectories and then we average over words to obtain the *mean anomalous diffusion exponent $\bar{\alpha}$*. Although individual trajectories deviate considerably from the simple scaling behavior given by $|\Delta x|^2 \sim t^\alpha$, somewhat surprisingly, their ensemble average, $\langle |\Delta x|^2(t) \rangle$, follows the scaling given by $t^{\langle \alpha \rangle}$ with high accuracy. This is illustrated in Fig 3a, showing the squared displacement $|\Delta x|^2(t)$ of several individual English words as well as the ensemble average $\langle |\Delta x|^2(t) \rangle$ of all English words.

The grey cells of Table 1 list obtained exponents $\langle \alpha \rangle$ and $\bar{\alpha}$ for five languages, English, French, German, Italian, and Spanish. In all languages, both the ensemble-average anomalous diffusion exponent $\langle \alpha \rangle$ and the mean anomalous diffusion exponent $\bar{\alpha}$ are significantly lower than $\alpha = 1$, i.e., semantic trajectories show subdiffusive behaviour on this time scale. In particular, we measured an ensemble-average anomalous diffusion exponent $0.4 < \langle \alpha \rangle < 0.5$ for all five languages. This is in strong contrast with a random walk generated using the same parameters (see Methods for details), corresponding to a fitted $\alpha \approx 1$, as shown by Fig 3b.

## Comparison with randomized trajectories

Given the robust observation that the ensemble-average semantic behaviour of words follows subdiffusion with $\langle \alpha \rangle \approx 0.4 - 0.5$ across languages, one might ask the following two questions. (i) Is this an artifact of the diachronic alignment procedure? (ii) If not, what is behind the observed subdiffusion? In other words, what (combination of) microscopic models of stochastic collective dynamics might explain this macroscopic result? In the following, we focus on these two questions. We apply a series of randomization methods to the original trajectories, gradually removing temporal correlations in step sizes and step directions of individual trajectories to see which of these, if any, play a role behind subdiffusion. As shown in Fig 3c and Table 1, step sizes of a trajectory are randomized three different ways, which we call *random sizes*, *sizes from distribution*, and *shuffled sizes*; step directions are randomized two ways,

*random directions*, *shuffled directions* (see Methods), which, together with the original trajectories, gives three times four combinations.

Fig 3c and 3d show the average squared displacement $\langle(\Delta x)^2\rangle$ of all English words, and the distribution of anomalous diffusion exponents $\alpha$ fitted to individual word trajectories separately, under all twelve combinations of trajectory randomization methods (including the original trajectories). The same plots for French, German, Italian, and Spanish are shown in S3 and S4 Figs in S1 File; fitted anomalous diffusion exponents $\langle\alpha\rangle$ and $\bar{\alpha}$ are listed for all five languages in Table 1. The top left panel (random step sizes, random step directions) corresponds to a random walk; the bottom right panel (original step sizes, original step directions) corresponds to the original, non-randomized semantic trajectories.

Comparing the results of original trajectories with the randomized trajectories (Fig 3c) conveys three important messages: (i) temporal alignment, applied to an ensemble of uncorrelated trajectories (top left panel), results $\langle\alpha\rangle = 1$ (see top left panel), equivalent to that of uncorrelated trajectories *without* alignment. (ii) Both correlations in step sizes and step directions are important factors behind subdiffusion. Neither of them alone can produce trajectories with an $\alpha$ lower than 0.8, yet the two combined give $\alpha \approx 0.5$. (iii) If step sizes do not vary significantly, the shuffling of the directions can alter the *total* displacement along a trajectory only by a small amount, thus, the last data points in the middle row (shuffled directions) must be very close to the last data points of the corresponding panels of the last row (original directions). On the other hand, shuffled directions do remove temporal correlations in step directions, making trajectories follow approximately diffusion-like behavior until the constraint on total displacement does not affect them. This can be clearly seen in the middle row of Fig 3c; in Fig 3d, we decided not to fit exponents to individual word trajectories in the middle row (shuffled directions) to avoid systematic bias in the exponents depending on the fitting range.

We further investigate a related effect, the effect of keeping the total size of the word *cloud* constant, as formalized by Eq (11). Ensembles of randomized trajectories in Fig 3c and 3d do not obey this constraint; we, therefore, generated a random walk, corresponding to random step directions and random step sizes, with the additional constraint on keeping the total cloud size constant. This simulated trajectory is illustrated in Fig 3b ("random walk"). While the total displacement is limited by the size of the word cloud, this only appears to affect the random walk trajectory when the displacement reaches the radius of the cloud (and then converges to approximately to the square root of two times the radius, corresponding to two orthogonal vectors with length equals to the radius). This, along with the middle row of Fig 3c, suggests that global constraints on total displacement do not cause the observed subdiffusive behavior.

Finally, we compare the original semantic trajectories to another randomized model, where time labels of the original data are randomly reshuffled (see details in Section S6 in S1 File). As shown by S5 Fig in S1 File, such randomized "trajectories" display a non-zero anomalous diffusion exponent, $\alpha \approx 0.2$, which can be attributed to the *alignment* of embeddings corresponding to subsequent times. This is an unavoidable effect whenever embedding dimensions are arbitrary and thus alignment is necessary; this baseline non-zero anomalous diffusion exponent is present even when time labels of diffusive trajectories ($\alpha = 1$) are reshuffled before the same pipeline, including alignment, is applied (S8 Fig in S1 File). Trajectories built from reshuffled time labels, however, display significantly larger fluctuations (S6 Fig in S1 File), and correspondingly, the goodness of fit of the anomalous diffusion exponent is significantly lower (S7 Fig in S1 File).

## Discussion

Cumulative culture is arguably one of the most distinct characteristic features of human behavior. Understanding "laws" that govern cultural evolution is a crucial step towards

understanding Homo Sapiens itself. Here we study cultural evolution through the evolution of *meaning* of words, as formalized by the distributional hypothesis: "a word is characterized by the company it keeps" [44]. Cultural evolution is notoriously difficult to study without making a large number of subjective assumptions and interpretations. One of the main contributions of this work is that it tries to make underlying assumptions (that are in the form of mathematical formalizations and their interpretation) as explicit as possible. We build on the work by Hamilton et al. [20] to find statistical regularities of *global* meaning change, i.e., with respect to a global semantic embedding, as opposed to a local semantic neighborhood, indicative of a stronger role of linguistic drift than cultural shift. We note that a classification of *local* stochastic trajectories (with respect to the semantic neighborhood) would provide a basis for a robust comparison of the above, a possible subject of future work.

Our data processing and analysis pipeline consists of three phases. First, semantic relations between all words are extracted through a state-of-the-art implementation of the distributional hypothesis: Word2vec, trained with the so-called Skip-gram with a negative sampling method. This algorithm provides a high-dimensional embedding of all words such that pairwise distances reflect semantic similarity. Apart from being fast and data-efficient, it is also well-anchored in human language representation through psycholinguistic studies. Second, to extract evolutionary trajectories, embeddings at different times need to be weaved together. This is a highly non-trivial process since the mapping between embedding and co-occurrence statistics is degenerate: many embeddings are consistent with the same co-occurrence data. When constructing trajectories, we need to break this symmetry: one specific embedding from each equivalence class needs to be chosen. We jointly construct equivalence classes and choose one specific representative of each class, both informed by the dynamics. In particular, we choose trajectories to be maximally smoothened. As we show, this maximal smoothening, applied to an ensemble of diffusing particles, does not alter the anomalous diffusion exponent $\alpha$ of the process, suggesting that it would not alter $\alpha$ significantly when actual semantic trajectories are considered, either. The third phase of the process is the comparison of the ensemble of semantic trajectories of words with various randomized counterpart ensembles. The randomization method we consider dissects various temporal correlations in semantic trajectories by randomizing step directions and step sizes separately, while still applying the same method for temporal alignment (i.e. trajectory smoothening).

With all this, we seek to answer the following questions: (i) is there any robust statistical regularity regarding the ensemble of actual semantic trajectories of words? (ii) If yes, what might be the reason? What microscopic dynamical rules can explain the observed macroscopic (ensemble-level) statistical regularities? This work provides an answer to the first question: semantic trajectories show different behaviour on the examined time scale from ordinary random walk (diffusion); semantic trajectories seem to be subdiffusive. In particular, on actual semantic trajectories extracted from data with our processing methodology we measure an anomalous diffusion exponent $\alpha \approx 0.4$ to 0.5, in strong contrast with a random walk belonging to the $\alpha = 1$ class. We note that the measured numerical value depends on the parameters of the process, especially the size of the applied sliding window. Since word embedding is a non-linear conversion of linguistic statistics to vector representation, it is not surprising that parameters introduced before that step have non-trivial effect on the result. The investigation of this dependence would require a deep analysis of the Word2vec algorithm, which goes beyond the scope of this paper, however we show in S1 File that this subdiffusive behavior appears with other parameter choices, and the approximate value of 0.4–0.5 is the upper limit where the exponent converges to. Hence the result of our analysis are in accordance with the classification of semantic drift as subdiffusion. This deviation from normal diffusion suggests that at least one of the criteria for diffusion mentioned in the Introduction must not hold. In

other words, semantic change cannot be entirely random; correlations must exist. Considering how human memory and collective knowledge influence language use, this might seem obvious. Still, the subdiffusive behavior offers a fresh perspective on the subject.

We point out here that short-range temporal correlations in trajectories, not extremely inhomogeneous step sizes, or weak correlations between trajectories do not cause the resulting trajectories to deviate from $\alpha = 1$. Instead, subdiffusion can be explained by qualitatively different microscopic dynamical rules. These include (i) long temporal correlations in trajectories, (ii) extremely inhomogeneous step size distributions, (iii) stochastic dynamics of "jamming" (overly densely packed) particles (here, words), (iv) changing average step sizes over time (corresponding to a changing diffusion coefficient), (v) diffusion in disordered media, and many others [27–30]. If one would like to describe the microscopic dynamics of the cognitive effects that can influence semantic change, the above observable statistical properties or some kind of combination of them should be obtained on an ensemble level. Although investigating possible combinations of these microscopic dynamics as explanations of semantic subdiffusion is a subject of future work, we can at least exclude some of them based on our results. Step size distribution ("Brownian vs Levy flight") and even correlations in step sizes do not seem to contribute to subdiffusion at all. Correlations in step directions explain some but far from all: ensembles of trajectories where step sizes are randomized but step directions are kept still exhibit $\alpha \approx 0.8$–$0.9$ in contrast with actual trajectories that follow $\alpha \approx 0.4$–$0.5$ (that include correlations both among step sizes and directions, and possibly cross-correlations between these categories too).

We highlight the following limitations of this analysis. First, semantic embedding methods, such as Word2Vec, compress pairwise statistical relationships between words by embedding them in a space with arbitrary dimensions. Consequently, embeddings corresponding to different times, such as subsequent years, have to be aligned to each other. This introduces an unavoidable bias in the properties of semantic trajectories; comparison with randomized models are necessary to distinguish artifacts of the pipeline from properties of the data. Second, semantic embedding and temporal alignment generates semantic trajectories of words jointly; the position of a word is determined by the position of all other words by minimizing a global cost function, suggesting that trajectories of individual words might be less informative than summary statistics of an ensemble of trajectories. Third, available data is highly unevenly distributed across time, necessitating a trade-off between the length of the trajectories and the embedding noise due to sparse data.

As in the biological examples where similar subdiffusive behavior arises, semantic change also raises questions both at a mechanistic, proximal level (what microscopic dynamical rules underlie subdiffusion?) and at an evolutionary, distal level (is subdiffusion adaptive or is it a consequence of physical-informational constraints? If it is adaptive, what is it *for* and what selection pressures led to its emergence?). In these cases, however, it is certainly known that the movement is governed by microscopic natural (physical or biological) laws which cause emergent macroscopic statistical effects. Even though such microscopic laws of language evolution are more context-dependent and noisy, and thus simple microscopic models provide a less powerful explanation to social and cognitive notions like semantic change or cultural memory, we indicate in this paper that their macroscopic (ensemble-level) behavior might converge to statistical classes, akin to universality classes that describe effective separation of scales in physics and computation.

## Supporting information

**S1 File. Supporting information.** We present and explain in this file all the auxiliary results and arguments that are related to this paper.
(PDF)

## Acknowledgments

The authors thank Douglas Moore, Yanbo Zhang, Jake Hanson, András Szántó, József Venczeli, and Péter Pollner for useful discussions at various stages of the project.

## Author Contributions

**Conceptualization:** Bogdán Asztalos, Gergely Palla, Dániel Czégel.

**Formal analysis:** Bogdán Asztalos, Gergely Palla, Dániel Czégel.

**Investigation:** Bogdán Asztalos, Gergely Palla, Dániel Czégel.

**Methodology:** Bogdán Asztalos, Gergely Palla, Dániel Czégel.

**Software:** Bogdán Asztalos.

**Supervision:** Gergely Palla, Dániel Czégel.

**Visualization:** Bogdán Asztalos.

**Writing – original draft:** Bogdán Asztalos, Gergely Palla, Dániel Czégel.

**Writing – review & editing:** Bogdán Asztalos, Gergely Palla, Dániel Czégel.

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
