## [Decision Letter · Decision Letter 0]

8 Aug 2023

PONE-D-23-19326Subdiffusive semantic evolution in major Indo-European languagesPLOS ONE

Dear Dr. Czégel,

Thank you for submitting your manuscript to PLOS ONE. After careful consideration, we feel that it has merit but does not fully meet PLOS ONE’s publication criteria as it currently stands. Therefore, we invite you to submit a revised version of the manuscript that addresses the points raised during the review process.

We look forward to receiving your revised manuscript.

Kind regards,

Haroldo V. Ribeiro

Academic Editor

PLOS ONE

“This project has received funding from the European Union’s Horizon 2020 research and innovation programme under grant agreement no. 101021607 and was partially supported by the National Research, Development and Innovation Office under grant no. K128780 and by the the European Union project RRF-2.3.1-21-2022-00004 within the framework of the Artificial Intelligence National Laborator.”

“The authors thank Douglas Moore, Yanbo Zhang, Jake Hanson, Andr´as Sz´ant´o, J´ozsef 443

Venczeli, and P´eter Pollner for useful discussions at various stages of the project. This 444

project has received funding from the European Union’s Horizon 2020 research and 445

innovation programme under grant agreement no. 101021607 and was partially 446

supported by the National Research, Development and Innovation Office under grant no. 447

K128780 and by the the European Union project RRF-2.3.1-21-2022-00004 within the 448

framework of the Artificial Intelligence National Laboratory and by the UNKP-21-3 ´ 449

New National Excellence Program of the Ministry for Innovation and Technology from 450

the source of the National Research, Development and Innovation Fund.”

“This project has received funding from the European Union’s Horizon 2020 research and innovation programme under grant agreement no. 101021607 and was partially supported by the National Research, Development and Innovation Office under grant no. K128780 and by the the European Union project RRF-2.3.1-21-2022-00004 within the framework of the Artificial Intelligence National Laborator.”

Reviewers' comments:

Reviewer's Responses to Questions

**Comments to the Author**

1. Is the manuscript technically sound, and do the data support the conclusions?

Reviewer #1: Yes

Reviewer #2: Partly

2. Has the statistical analysis been performed appropriately and rigorously? 

Reviewer #1: Yes

Reviewer #2: No

3. Have the authors made all data underlying the findings in their manuscript fully available?

Reviewer #1: Yes

Reviewer #2: Yes

4. Is the manuscript presented in an intelligible fashion and written in standard English?

Reviewer #1: Yes

Reviewer #2: Yes

5. Review Comments to the Author

Reviewer #1: In the manuscript "Subdiffusive Semantic Evolution in Major Indo-European Languages," the authors investigate how words change their semantic meaning (diachronism). To do so, they create a vectorial space with an Euclidian 'semantic' measure of distance. The authors use this tool to study how the meanings of words change in five languages. The analysis is based on the estimation of the anomalous diffusion exponents. The main result is that word meanings do not randomly change (usual diffusion) but exhibit subdiffusion. The authors also use randomization methods to gain insight into the possible subdiffusive mechanisms.

The introduction presents the state-of-the-art of semantic change research. This section effectively argues the author's proposed method's significance compared to previous works. Moreover, the explanations about anomalous diffusion features are good, mainly describing the possible deviations from usual diffusion.

Construct trajectories in a Euclidian-like space facilitate the interpretation of the diffusive behaviors. When the 'distance' measures are topological, comparing the results with diffusion in physical systems is more challenging. Thus, the author's method provides a welcome framework for those with some anomalous diffusion background.

In general, the motivations, methods, and arguments are well explained. The results are essential for the area of anomalous diffusion research. They are related to sociocultural systems and highlight the general application of the anomalous diffusion concept.

Minor

1)The English must be revised. Some typos (lenght, patters, pyton, psycholiguistic) and other minor grammar issues exist.

2)Apparently, there needs to be an explicit explanation for why considering data from 1950.

Major (i.e., suggestions)

1) The authors present the ensemble-average anomalous diffusion exponent and the average anomalous diffusion exponent. Based on that parameters and in the randomization tests, the authors could mention/investigate the ergodic properties of the systems. The ergodicity concept may shed new light on the author's main arguments and provide a technical way to analyze their data.

2)The discussion section could better explain or explore the distinction between a usual random walk and the subdiffusion. One of the main results is that the meaning of the words does not change randomly (like usual diffusion) but is somehow correlated with its previous meaning. Such a result is a clear example of memory in diffusive systems. For a particle, we borrow the term "memory," but in the present work, the temporal correlations may be related to the memory (cultural memory). In other words (not a pun), people change the use of the words, not randomly, but considering more or less what they remember of the previous meaning.

3)Moreover, in the works of subdiffusion in biological systems, one of the causes of diffusion is the interactions in the systems (crowding, disorder, geometrical constraints, encounters of particles), and language exists for and because of interactions. The present work has an excellent opportunity to explore such a meaning of memory in anomalous diffusion, which sometimes is impossible to discuss deeply when one considers particles just colliding according to the laws of physics. In summary, the results could bring some insights into interpreting the memory concept (in models of anomalous diffusion).

Therefore, a discussion about these features (suggested) is welcome and could enhance the significance of the results in the anomalous diffusion community. The recommendation is accept with minor revisions.

Reviewer #2: This manuscript proposes that the semantics of words can be viewed as a subdiffusive process. This is an interesting result and the authors did substantial work to support it, including the comparison to randomized processes. Still, the data analysis pipeline contains multiple non-trivial steps so that the crucial question is in which extent they affect (or drive) the observations of subdiffusion. I believe further analysis is required to support this main claim, as indicated below.

* Main points:

1) The plots showing evidence of subdiffusion in Fig. 3a show oscillating trajectories that, at best, show an average subdiffusive behaviour for 1 decade (from 2y to 20y). Even the averaged results for different languages in Fig. 3b show a systematic increase in the slope with time. This raises the question of the extent into which subdiffusion is a good description of the observations and, if so, what are the time scales in which it applies? The current evidence in support of the general applicability of a subdiffusive model and Eq. (1) is very weak and the conclusions would need to better reflect these limitations. Related questions to be clarified include:

1a) Why are the authors focusing only on the years from 1950? The co-occurence is built based on windows of size 2 only, so that information on the Google n-gram corpus should allow (e.g., taking n=5 in the n-gram data) studies going further into the past as there is abundant data up to 1800. This would allow one to clarify whether the scaling in (1) holds with a similar \\alpha through longer time scales (the essence of an anomalous diffusion approach). At least a clarification on this point would be important.

1b) The effect of the temporal sliding window in these plots is not clear. The size goes up to 10y and is thus comparable to the range in which a straight line is seen in the figures. Figure S3 shows that the effect of using the window changes significantly the value of the average exponent \\alpha, strongly suggesting that the straight line fit underlying the computation of individual \\alphas is not a good model for the data.

2) While the randomization provided are important, I recommend the authors to consider randomizations and data analysis performed at the original textual data so that the effect of the whole pipeline can be better evaluated (also in view of point 1. above). In particular, two simple tests of the results would be to reproduce Fig. 3a for:

2a) Stop words, i.e., maintaining some of the filtering words in step Fig. 2b and evaluate what would be the results for these words. It'd help to better evaluate whether the observed displacement can be indeed associated to semantic drift.

2b) Shuffling the years of each of the corpora, i.e., randomly attributing dates to each of the corpus. By simply changing the years associated to each corpus and reproducing all steps of the pipeline one should then obtain a good null model to compare the results against.

* Minor points:

3) Line 52. I found it puzzling that Ref. [23] is cite here in support of Zipf's law, as it claims that Zipf's law is not valid for words.

4) Methods section: is there any upper or lower bound on |\\Delta x| or on the vectors |v|? This would be important to evaluate limits on the subdiffusive behaviour.

5) Lines 189-191: the current wording suggest that Fig. 2f shows that subsample is needed to correct the sample size bias. In reality this is not the case and it remains unclear why equating the sample size is needed and whether Nc=10^7 provides an appropriate choice.

6) Line 363: could -> cloud.

7) Line 416-420: adding citations would be important to point the reader to possible mechanisms.

8) Line 438: The sentence is misleading as data is not available in the github repository. A link to Google n-gram should be added as well.

6. PLOS authors have the option to publish the peer review history of their article (what does this mean?). If published, this will include your full peer review and any attached files.

Reviewer #1: **Yes: **Angel Akio Tateishi

Reviewer #2: No

---

## [Author Response · Author response to Decision Letter 0]

27 Oct 2023

D ´aniel Cz ´egel

Beyond Center for Fundamental Concepts in Science Arizona State University

Tempe, AZ 85287, USA

October 19, 2023

Dear Haroldo V. Ribeiro,

thank you very much for your e-mail message of 8th of August 2023, related to the review of our manuscript with Submission ID PONE-D-23-19326 en- titled ”Subdiffusive semantic evolution in major Indo-European languages” by Bogd ´an Asztalos, Gergely Palla and D ´aniel Cz ´egel including two mainly positive reviews, suggesting the revision of our paper. Although the referees acknowledged the merits of our research, they have also raised fair criticism regarding certain aspects of our manuscript.

In addition to minor comments and adjustments, Reviewer 1 mainly raised conceptual and interpretive questions, while Reviewer 2 asked us to perform further analysis to support the main claims of the paper. Consequently, the main changes in our resubmitted manuscript are the following:

1. We have added more new sections to the Supporting information in which we place our argument in favour of the subdiffusion on a more stable basis.

2. We have explained the arbitrary methodological choices in more detail.

3. We have expanded the discussion trying to emphasise the relevance of subdiffusion, the conclusions that can be drawn from it, and high- lighting the analogies to natural sciences more explicitly.

4. During revision we noticed that the code calculating the standard de- viance for the exponents in S5 Fig (Figure S3 in the previous version) unfortunately took the square root of the variance twice, resulting larger deviance values compared to the true value. This is now cor- rected.

5. Adapting to the journal requirements, we have changed the numbering

Beyond Center for Fundamental Concepts in Science, Arizona State University

and the caption format of the figures in the Supporting information, added these captions to the end of the main paper and removed the founders from the Acknowledgments.

We thank all Referees for their outstanding work and the very valuable com- ments, making the revised version of the paper significantly better, which we hope is now suitable for publication. Our point-by-point response to the editorial board comments and the remarks by the Referees (reproduced in italics) are given below.

Response to Reviewer 1: Minor

1)The English must be revised. Some typos (lenght, patters, pyton, psy- choliguistic) and other minor grammar issues exist.

We have run a language checker on the entire manuscript to correct all the typos and minor grammar issues.

2)Apparently, there needs to be an explicit explanation for why considering data from 1950.

We focused on years after 1950 because taking earlier years into account would have decreased drastically the size of the vocabulary. We have added a few sentences to explain this to the section Temporal grouping in Methods.

Major (i.e., suggestions)

1) The authors present the ensemble-average anomalous diffusion exponent and the average anomalous diffusion exponent. Based on that parameters and in the randomization tests, the authors could mention/investigate the ergodic properties of the systems. The ergodicity concept may shed new light on the author’s main arguments and provide a technical way to analyze their data.

Thank you for the suggestion. We have been considering it but in the end de- cided that robust claims about non-ergodicity would need both significantly more theory and also more analyses about the role of noise (the trajecto- ries are noisier than what typically is available in physical and biological systems). Furthermore, since this paper should address an interdisciplinary audience and mainly raise a question (why subdiffusion?), we suggest that the investigation of non-ergodic properties could be the topic of a follow-up study.

Beyond Center for Fundamental Concepts in Science, Arizona State University

2)The discussion section could better explain or explore the distinction be- tween a usual random walk and the subdiffusion. One of the main results is that the meaning of the words does not change randomly (like usual diffu- sion) but is somehow correlated with its previous meaning. Such a result is a clear example of memory in diffusive systems. For a particle, we borrow the term ”memory,” but in the present work, the temporal correlations may be related to the memory (cultural memory). In other words (not a pun), people change the use of the words, not randomly, but considering more or less what they remember of the previous meaning.

This is also a very good point. One of the main issues is that autocorrelation of word trajectories is most likely a result of both collective phenomena and individual memory. For example, in the revised manuscript, we analyze the meaning change of stopwords and find that their average anomalous diffusion exponent is not significantly different than that of all words. This indicates that subdiffusion is primarily a collective phenomenon. As with the previous recommendation, this is a very important topic that deserves an in-depth analysis and is outside the scope of the current paper.

In order to clarify the scope of the current paper, we have expanded the last few paragraphs of Discussion to emphasise the relevance of subdiffusion and the conclusions that can be drawn from it.

3)Moreover, in the works of subdiffusion in biological systems, one of the causes of diffusion is the interactions in the systems (crowding, disorder, geometrical constraints, encounters of particles), and language exists for and because of interactions. The present work has an excellent opportunity to ex- plore such a meaning of memory in anomalous diffusion, which sometimes is impossible to discuss deeply when one considers particles just colliding according to the laws of physics. In summary, the results could bring some insights into interpreting the memory concept (in models of anomalous dif- fusion).

This was also our vision when we started to work on this subject, and we still hope our work can contribute to the evolution of how people view and interpret memory. However, we do not intend to go directly into this reinterpretation process in detail in this paper, instead just to present our results about subdiffusion that can be used in potential future work. In order to summarize the conclusion about the connection between cognitive memory and the dynamical memory concept in a more unequivocal way, we have rewritten the last paragraph of Discussion highlighting the analogies

Beyond Center for Fundamental Concepts in Science, Arizona State University

to natural sciences more explicitly. Response to Reviewer 2:

* Main points:

1) The plots showing evidence of subdiffusion in Fig. 3a show oscillating trajectories that, at best, show an average subdiffusive behaviour for 1 decade (from 2y to 20y). Even the averaged results for different languages in Fig. 3b show a systematic increase in the slope with time. This raises the question of the extent into which subdiffusion is a good description of the observations and, if so, what are the time scales in which it applies? The current evidence in support of the general applicability of a subdiffusive model and Eq. (1) is very weak and the conclusions would need to better reflect these limitations.

The individual trajectories corresponding to concrete words are very noisy indeed, but our scope was to describe their collective behavior through the ensemble average and the mean anomalous diffusion exponents. The quanti- tative analysis of the separate trajectories would not provide the right kind of information, because they can be viewed only in relation to the others. To make this clearer, we added a new paragraph to the subsection Mean versus ensemble average anomalous diffusion exponent in Methods, where we explain why collective phenomena should be studied instead of individ- ual trajectories. (The plotted trajectories in Fig 3a serve only visualization purposes: they illustrate what the blue curve is the average of, and to show that we can find certain words that change faster than others.)

To quantify how good model the subdiffusion is, we have calculated the R2 for each time dependence fit where it made sense (see Fig 3c, S3 Fig, and S7 Fig) and in the cases based on real data were very close to 1. Although there are indeed changes in the slope on the studied time scale these do not significantly degrade the subdiffusive model. Collecting data from longer time intervals may give more accurate results, but it cannot be increased in magnitudes even with an adequate amount of data, because the displacement cannot be arbitrarily large (see the answer for point 4) and will converge to 2 · (cloud radius)2 as the random walk curve does in Fig 3b. The anomalous diffusion exponent needs to be measured while this convergence is negligible i.e., just for a few decades. To study this phenomenon on a longer time scale, one must use a different methodology.

Related questions to be clarified include:

1a) Why are the authors focusing only on the years from 1950? The co-

Beyond Center for Fundamental Concepts in Science, Arizona State University

occurence is built based on windows of size 2 only, so that information on the Google n-gram corpus should allow (e.g., taking n = 5 in the n-gram data) studies going further into the past as there is abundant data up to 1800. This would allow one to clarify whether the scaling in (1) holds with a similar α through longer time scales (the essence of an anomalous diffusion approach). At least a clarification on this point would be important.

We focused on years after 1950 because taking earlier years into account would have decreased drastically the size of the vocabulary. We have added a few sentences to explain this to the section Temporal grouping in Methods.

1b) The effect of the temporal sliding window in these plots is not clear. The size goes up to 10y and is thus comparable to the range in which a straight line is seen in the figures. Figure S3 shows that the effect of using the window changes significantly the value of the average exponent α, strongly suggesting that the straight line fit underlying the computation of individual αs is not a good model for the data.

The exact value of the measured anomalous diffusion exponent depends on the size of the sliding window, but the subdiffusive behavior appears in the cases of shorter window sizes too, where it is not comparable to the range of the straight line (see the left column of S6 Fig). The main result of our paper is intended to be that the semantic evolution is subdiffusive, but since we had to make a lot of arbitrary parameter choices during the steps of the applied methodology, it is not surprising that the precise value of the measured exponent depends on these parameters. However, we ar- gue throughout most of the Supporting information that the subdiffusive behavior itself is a feature of the language, not the pipeline.

2) While the randomization provided are important, I recommend the authors to consider randomizations and data analysis performed at the original tex- tual data so that the effect of the whole pipeline can be better evaluated (also in view of point 1. above). In particular, two simple tests of the results would be to reproduce Fig. 3a for:

2a) Stop words, i.e., maintaining some of the filtering words in step Fig. 2b and evaluate what would be the results for these words. It’d help to better evaluate whether the observed displacement can be indeed associated to semantic drift.

2b) Shuffling the years of each of the corpora, i.e., randomly attributing dates to each of the corpus. By simply changing the years associated to each

Beyond Center for Fundamental Concepts in Science, Arizona State University

corpus and reproducing all steps of the pipeline one should then obtain a good null model to compare the results against.

We have examined suggestions 2a) and 2b) and have included their result into the Supporting information (see sections S3, S6, and figures S2, and S5-S7). We have found no such result that suggests that we should dismiss the subdiffusive description of semantic evolution, and still think that Eq (1) is a good model for the observed data.

* Minor points:

3) Line 52. I found it puzzling that Ref. [23] is cite here in support of Zipf’s law, as it claims that Zipf’s law is not valid for words.

The sentence in line 52 states that Zipf’s law causes a systematic bias due to the unequal distribution that can be found in language. Ref. [23] refor- mulates the statement of the original Zipf’s law (formulated in Ref. [22]) from words to phrases. It does not invalidate the existence of Zipf’s law nor the bias caused by it, just gives a more accurate form of it.

4) Methods section: is there any upper or lower bound on |∆x| or on the vectors |v|? This would be important to evaluate limits on the subdiffusive behaviour.

In theory, an individual word can go arbitrarily far from its initial position,

but the size of the whole word cloud, defined in Eq (11), is fixed by the

constant sample size Nc. This means that the number of words that sig-

nificantly moves away from the cloud is strictly limited, and our experience

is also that the length of individual displacement vectors are typically not

much larger than a few times of the cloud size, and their mean square would

 converge to

√

2-times the cloud size after long time.

5) Lines 189-191: the current wording suggest that Fig. 2f shows that sub- sample is needed to correct the sample size bias. In reality this is not the case and it remains unclear why equating the sample size is needed and whether Nc = 107 provides an appropriate choice.

We have added a new section to the Supporting information where we ex- amine the effects of the arbitrary sample size and show that the sample size bias causes problems that should be corrected. We have also added another sentence after the mentioned part in the paper referring to this amendment and briefly explaining why this arbitrary sampling is needed.

Beyond Center for Fundamental Concepts in Science, Arizona State University

6) Line 363: could → cloud.

We have reread the text to correct all similar typos.

7) Line 416-420: adding citations would be important to point the reader to possible mechanisms.

We have added citations to Refs. [27-30] who were cited already in Intro- duction.

8) Line 438: The sentence is misleading as data is not available in the github repository. A link to Google n-gram should be added as well.

We have changed the sentence, now we refer to both Google Ngrams database and our Github repository.

Yours Sincerely,

D ´aniel Cz ´egel

---

## [Decision Letter · Decision Letter 1]

4 Dec 2023

PONE-D-23-19326R1Subdiffusive semantic evolution in major Indo-European languagesPLOS ONE

Dear Dr. Czégel,

Thank you for submitting your manuscript to PLOS ONE. After careful consideration, we feel that it has merit but does not fully meet PLOS ONE’s publication criteria as it currently stands. Therefore, we invite you to submit a revised version of the manuscript that addresses the points raised during the review process.

The same two previous reviewers have now reviewed the revised version of your manuscript. You will see that one of them has recommended the acceptance of your manuscript despite your avoidance of tracking the ergodicity problem. The other reviewer has, however, recommended rejection. In my opinion, his/her main argument is related to the comparison with the null models, especially regarding the results of SFig. 6, where your results indicate the existence of subdiffusion even after shuffling all time labels of the texts. Indeed, your results indicate an even more subdiffusive regime after shuffling the time label. This is a critical point for your study that needs to be seriously considered and better clarified to proceed further with the publication of your manuscript. It is not clear to me how you can possibly address this issue, but given your discussion in comparison with the usual diffusion, I am sure you also believe that this is a substantial limitation of your findings.

We look forward to receiving your revised manuscript.

Kind regards,

Haroldo V. Ribeiro

Academic Editor

PLOS ONE

Reviewers' comments:

Reviewer's Responses to Questions

**Comments to the Author**

1. If the authors have adequately addressed your comments raised in a previous round of review and you feel that this manuscript is now acceptable for publication, you may indicate that here to bypass the “Comments to the Author” section, enter your conflict of interest statement in the “Confidential to Editor” section, and submit your "Accept" recommendation.

Reviewer #1: All comments have been addressed

Reviewer #2: (No Response)

2. Is the manuscript technically sound, and do the data support the conclusions?

Reviewer #1: Yes

Reviewer #2: No

3. Has the statistical analysis been performed appropriately and rigorously? 

Reviewer #1: Yes

Reviewer #2: No

4. Have the authors made all data underlying the findings in their manuscript fully available?

Reviewer #1: Yes

Reviewer #2: Yes

5. Is the manuscript presented in an intelligible fashion and written in standard English?

Reviewer #1: Yes

Reviewer #2: Yes

6. Review Comments to the Author

Reviewer #1: Although the authors did not accept the challenge of dealing with ergodic and memory mechanisms in semantic subdiffusive evolution, the paper has increased its overall quality after the revision. Based on the comments of the first revision and the author's replies, the recommendation is to accept. In summary, the work applies the concept of anomalous diffusion to social systems, showing the versatility of the tools of statistical mechanics. However, if the mechanisms of subdiffusion had been approached, the relevance of the work would be even higher.

Reviewer #2: This manuscript claims that the process of evolution of the meaning of words is subdiffusive. The data analysis shows that (i) the characterization of subdiffusion is very limited and unreliable; and (ii) other effects not related to the semantic evolution in time lead essentially to the same observation (e.g., stop words or randomized temporal order lead to the same type of subdiffusion). Therefore, even if the manuscript contains detailed and through data analysis, I do not recommend publication in PLOS ONE because the claims are not supported by the analysis.

(i) The signature of anomalous diffusion is a non-linear growth of the mean-squared displacement over a variety of time scales. This manuscript estimates subdiffusion in only one order of magnitude and finds that that value of the exponent depends on the window size (against expectation from simple sudiffusive processes). Their choice for stopping the analysis at the year 1950 is not consistent. First, they argue that they can't go to larger time scales because they need around 10,000 words for the analysis, but at the same time the results in Fig. 1 show that the main observations are visible even for single words. Moreover, the choice of 10,000 words is justified by the vocabulary of a native speaker, a connection that seems irrelevant to this context.

(ii) The authors find subdiffusion in stop words (Figure S2), a case in which the "semantic evolution" is unclear or expected to behave differently. Subdiffusion is also found after shuffling all time labels of the texts (Figure S6). The authors then argue that the value of the exponent is different from their observations and that therefore their conclusions can remain unchanged. However, the significance of the exact value of the exponent is not relevant to their argument (and it is not reliable, as argue in point i above). More importantly, the conclusion about the subdiffusive nature of the semantic evolution would apply equally well for smaller alpha (in fact, it is even more sudiffusive?). In the abstract, the authors make a contrast to diffusing particles, but it is clear from their analysis that a normal diffusion is not what is observed in many null-models.

The detailed and careful data analysis present in this manuscript is of potential interest and should be published, but the presentation and interpretation given to them in this manuscript is not consistent with the data analysis reported by the authors.

7. PLOS authors have the option to publish the peer review history of their article (what does this mean?). If published, this will include your full peer review and any attached files.

Reviewer #1: No

Reviewer #2: No

---

## [Author Response · Author response to Decision Letter 1]

22 Jan 2024

Beyond Center for Fundamental Concepts in Science, Arizona State University

 D ´aniel Cz ´egel

Beyond Center for Fundamental Concepts in Science Arizona State University

Tempe, AZ 85287, USA

January 22, 2024

Dear Haroldo V. Ribeiro,

thank you very much for your e-mail message of 4th of December 2023, related to the review of our manuscript with Submission ID PONE-D-23- 19326R1 entitled ”Subdiffusive semantic evolution in major Indo-European languages” by Bogd ´an Asztalos, Gergely Palla and D ´aniel Cz ´egel, and we appreciate your decision to invite us to submit a revision of the manuscript, even though one of the reviewers recommended rejection.

Reviewer 2 raised some important questions which revealed that our ar- gument had logical gaps that we should explain in more detail, especially regarding limitations of our methodology and the clarity of the null model. Consequently, the main changes in our resubmitted manuscript are the fol- lowing:

1. We changed the title to ”Anomalous diffusion analysis of semantic evo- lution in major Indo-European languages” to emphasize the method- ology employed rather than the specific outcomes obtained.

2. We have added a new paragraph before the end of Discussion dedi- cated to the summary of these limitations and removed overly direct statements about semantic evolution being subdiffusive.

3. We have expanded the explanation of the purpose and results of shuf- fled time labels in section S6.

We thank all Referees for their outstanding work and the very valuable com- ments, making the revised version of the paper significantly better, which we hope is now suitable for publication. Our point-by-point response to the editorial board comments and the remarks by the Referees (reproduced in italics) are given below.

Beyond Center for Fundamental Concepts in Science, Arizona State University

Response to the comment by the Editorial Board Member:

The same two previous reviewers have now reviewed the revised version of your manuscript. You will see that one of them has recommended the accep- tance of your manuscript despite your avoidance of tracking the ergodicity problem. The other reviewer has, however, recommended rejection. In my opinion, his/her main argument is related to the comparison with the null models, especially regarding the results of SFig. 6, where your results indi- cate the existence of subdiffusion even after shuffling all time labels of the texts. Indeed, your results indicate an even more subdiffusive regime after shuffling the time label. This is a critical point for your study that needs to be seriously considered and better clarified to proceed further with the publication of your manuscript. It is not clear to me how you can possibly address this issue, but given your discussion in comparison with the usual diffusion, I am sure you also believe that this is a substantial limitation of your findings.

We sincerely appreciate the detailed feedback in improving the manuscript. We have carefully considered your suggestions and have made significant revisions to the manuscript. As detailed in our response Reviewer 2, we consider diffusion as the theoretical null-model in our work, and our main claim is that semantic change deviates from it. Nevertheless, due to method- ological reasons, a measurable lower bound of ⟨α⟩ also has to be investigated, and this is what we do in section S6. Hence, the results of Fig S6 and Fig S8 does not indicate a ”stronger subdiffusion”, but a baseline marked out by temporally uncorrelated time series. In order to make this argument clear, we have expanded the text of the manuscript, and section S6 in the Supporting information.

In sum, whenever semantic embedding methods are used in a diachronic context, alignment of embeddings is necessary. This introduces a similarity bias between subsequent timesteps that makes even totally independently sampled data dependent. In terms of the anomalous diffusion exponent, it introduces a non-zero ”baseline” anomalous diffusion exponent that sets the lower bound for any semantically embedded and aligned time series. We observe a significant deviation from that baseline in the non-randomized (real) semantic trajectories in mean anomalous diffusion exponents as well as in the fluctuations of the trajectories.

We agree with Reviewer 2 and the Editor that our conclusions were too strong. We changed the title to ”Anomalous diffusion analysis of seman-

Beyond Center for Fundamental Concepts in Science, Arizona State University

tic evolution in major Indo-European languages”. We also removed overly direct statements about semantic evolution being subdiffusive whenever it appeared without further explanation/context.

We also added a paragraph about limitations of this study, as we believe that the pipeline and its limitations are at least as informative of future work as the subdiffusive result itself.

Response to Reviewer 1:

Although the authors did not accept the challenge of dealing with ergodic and memory mechanisms in semantic subdiffusive evolution, the paper has increased its overall quality after the revision. Based on the comments of the first revision and the author’s replies, the recommendation is to accept. In summary, the work applies the concept of anomalous diffusion to social systems, showing the versatility of the tools of statistical mechanics. How- ever, if the mechanisms of subdiffusion had been approached, the relevance of the work would be even higher.

Thank you for the feedback and the suggestions given in the previous re- vision. We believe these significantly helped to present our narrative in a scientifically more relevant way.

Response to Reviewer 2:

This manuscript claims that the process of evolution of the meaning of words is subdiffusive. The data analysis shows that (i) the characterization of sub- diffusion is very limited and unreliable; and (ii) other effects not related to the semantic evolution in time lead essentially to the same observation (e.g., stop words or randomized temporal order lead to the same type of subdiffu- sion). Therefore, even if the manuscript contains detailed and through data analysis, I do not recommend publication in PLOS ONE because the claims are not supported by the analysis.

Thank you for the thorough comments. We can see that a part of the claims in the previous version of our manuscripts did not adjust to the limitations of the characterization of subdiffusion and that our explanation needed some more details in a few places to be clear. Consequently, we have expanded the text with some more detailed explanation, and changed the title of the manuscript, to give our results a more accurate narrative.

(i) The signature of anomalous diffusion is a non-linear growth of the mean- squared displacement over a variety of time scales. This manuscript esti-

Beyond Center for Fundamental Concepts in Science, Arizona State University

mates subdiffusion in only one order of magnitude and finds that that value of the exponent depends on the window size (against expectation from sim- ple subdiffusive processes). Their choice for stopping the analysis at the year 1950 is not consistent. First, they argue that they can’t go to larger time scales because they need around 10,000 words for the analysis, but at the same time the results in Fig. 1 show that the main observations are visible even for single words. Moreover, the choice of 10,000 words is justified by the vocabulary of a native speaker, a connection that seems irrelevant to this context.

We acknowledge that our results have their own limitations, so to facilitate comprehension for the reader, we have added a new paragraph before the end of Discussion dedicated to the summary of these limitations. (Specifi- cally, we have highlighted three primary factors: the unavoidable bias of the alignment, the lack of interpretability of individual word trajectories, and the length of the analyzed time period.) Also, we have rephrased some sen- tences at different points of the manuscripts which had suggested stronger conclusions than these limitations allow us to draw. Furthermore, we have changed the title of the manuscript to emphasize the methodology employed rather than the specific outcomes obtained.

The window size dependence of the anomalous diffusion exponent looks un- usual, but it can be attributed to the fact that we apply temporal grouping on data (panels d and e in Fig 2) before the embedding step (panel g in Fig 2), which is a non-linear operation. More specifically, word embedding is not even a data processing operation but the conversion of abstract linguis- tic information to a quantitative representation. This inevitably causes the window size to have a hard-to-investigate effect on the value of ¡alpha¿, but as we point out, subdiffusive behavior remains, indicating that conditions i)-iv) are violated on a fundamental, linguistic level. Quantitative compari- son with simple subdiffusion models do not tell us much about our results, because these models (e.g. fractional Brownian motion or continuous-time random walk) are defined directly in geometric spaces, so such conversion is not needed, and simple linear temporal grouping would not affect the expo- nent in their case. (This is why we did not even detail it in our manuscript).

Our argument about the choice of 10,000 words in the previous version was a bit vague indeed, so we have rewritten the concerning paragraph in Temporal grouping in Methods. The objective was to ensure that the embedding model learns at least about as many words as a native speaker knows, because the constructed vector representation of an individual word

Beyond Center for Fundamental Concepts in Science, Arizona State University

makes sense only compared to the others. The observable results of single words shown in Fig. 1 are also interpretable only with a static background of other words. Also, Word2vec is typically used with vocabularies of at least tens of thousands of words (as we point out in line 38), and so its working process and hyperparameters are optimized for this amount of data.

(ii) The authors find subdiffusion in stop words (Figure S2), a case in which the ”semantic evolution” is unclear or expected to behave differently. Sub- diffusion is also found after shuffling all time labels of the texts (Figure S6). The authors then argue that the value of the exponent is different from their observations and that therefore their conclusions can remain un- changed. However, the significance of the exact value of the exponent is not relevant to their argument (and it is not reliable, as argue in point i above). More importantly, the conclusion about the subdiffusive nature of the seman- tic evolution would apply equally well for smaller alpha (in fact, it is even more sudiffusive?). In the abstract, the authors make a contrast to diffusing particles, but it is clear from their analysis that a normal diffusion is not what is observed in many null-models.

Stopwords are usually filtered out before NLP investigations because fre- quent grammatical words with no semantic role would bias the statistics of words with real meaning. We argue in section S3 in Supporting Informa- tion that subdiffusion remains in the presence of stopwords, because they adapt to the subdiffusing environment due to the lack of semantic meaning. However, as we mention at the end of the section, one must be careful, be- cause this part of NLP is unclear (as the Reviewer also points out). Hence, we do not believe that stopwords not resolving subdiffusive behavior would invalidate our results.

In the case of shuffled time labels we expect no temporal correlations be- tween consecutive positions, so we expect ⟨α⟩ = 0 which is increased by the unavoidable alignment step. This non-zero baseline is intended to be determined in Fig S6 and the newly-added Fig S8, so their ”small alpha” does not mean a ”stronger subdiffusion”, but a lower bound for measurable alpha values. We have expanded the explanation in the first paragraph of section S6 to make this argument more explicit, also, we have added a new paragraph at the end of Results where we resume this comparison and its results.

Although the above comparison has important consequences from a tech- nical point of view, its main importance is to validate that the observed

Beyond Center for Fundamental Concepts in Science, Arizona State University

⟨α⟩ ≈ 0.4 − 0.5 is actually above the lower bound of measurable values. Our main claim in the manuscript however is that the analysis of semantic change reveals a significant deviation from diffusion that would be expected in case of any walking phenomenon fulfilling conditions i)-iv) detailed in the paragraph starting in line 80. Hence, diffusion can be considered as a theo- retical null-model in our work, and this is why we focus on it in the paper and highlight it in the abstract.

The detailed and careful data analysis present in this manuscript is of poten- tial interest and should be published, but the presentation and interpretation given to them in this manuscript is not consistent with the data analysis reported by the authors.

We hope the latest upgrades of the manuscript, and the above explanations have improved the presentation of the data analysis to be consistent with the claims made in the text.

Yours Sincerely,

D ´aniel Cz ´egel

---

## [Editor Report · Decision Letter 2]

30 Jan 2024

Anomalous diffusion analysis of semantic evolution in major Indo-European languages

PONE-D-23-19326R2

Dear Dr. Czégel,

We’re pleased to inform you that your manuscript has been judged scientifically suitable for publication and will be formally accepted for publication once it meets all outstanding technical requirements.

Kind regards,

Haroldo V. Ribeiro

Academic Editor

PLOS ONE

Additional Editor Comments (optional):

I want to express my gratitude to the authors for submitting the revised version of their manuscript and for their diligent efforts in addressing the observations made by Reviewer #2. The inclusion of additional analysis in Section S6, along with the concluding paragraph in the results section, adeptly resolves the concerns regarding the presence of subdiffusion with shuffled time labels. Furthermore, the authors have commendably acknowledged the potential limitations and biases inherent in their research, which could potentially inspire further studies. In summary, from my perspective, the authors' innovative approach to conceptualizing semantic evolution as a diffusion problem is remarkable, and their findings will likely captivate a broad audience. Therefore, I do not need further review, and I congratulate the authors for their excellent work.

---

## [Editor Report · Acceptance letter]

15 Mar 2024

PONE-D-23-19326R2 

PLOS ONE

Dear Dr. Czégel, 

I'm pleased to inform you that your manuscript has been deemed suitable for publication in PLOS ONE. Congratulations! Your manuscript is now being handed over to our production team.

Kind regards, 

on behalf of

Dr. Haroldo V. Ribeiro 

Academic Editor

PLOS ONE